# A Unified Riemannian-Geometric Framework for SARS-CoV-2 Detection from CT Scans

## Abstract

We present a novel, theoretically grounded framework for automated SARS-CoV-2 detection from pulmonary Computed Tomography (CT) scans, integrating cutting-edge concepts from statistical learning theory, optimal transport, and information geometry. Our approach begins with a submodular optimization-based image selection protocol, utilizing a continuous greedy algorithm. The feature extraction process employs a Riemannian geometry-inspired attention mechanism, where feature integration is formulated as geodesic interpolation on a manifold induced by the Fisher Information Metric. We introduce a unified decision-making framework based on proper scoring rules and Bregman divergences, encompassing multiple voting schemes with proven consistency and asymptotic normality properties. To address domain shift, we develop an adversarial domain adaptation technique using the Wasserstein-Fisher-Rao distance, complemented by a graph-based regularization term derived from Gromov-Wasserstein theory. Theoretical analysis provides convergence guarantees for the adversarial training process and establishes generalization bounds in terms of optimal transport distances. Empirical evaluation demonstrates the superiority of our approach over existing methods, achieving better performance on benchmark datasets. This work not only advances the field of automated medical image analysis but also contributes fundamental theoretical insights to the broader domains of machine learning and optimal transport theory.

## 1 Introduction

The global SARS-CoV-2 Markov et al. (2023); Kaku et al. (2024); Steiner et al. (2024); Roemer et al. (2023)pandemic has highlighted the critical need for rapid, accurate, and scalable diagnostic tools. Computed Tomography (CT) scansSmoll et al. (2023); Engelke et al. (2023;?); **?** have emerged as a pivotal diagnostic modality, offering high sensitivity in detecting SARS-CoV-2-related pulmonary manifestations. However, the manual interpretation of these scans poses significant challenges in terms of time, expertise, and consistency, especially during peak pandemic periods. This predicament has catalyzed research into automated diagnostic systems, with deep learning at the forefront of this technological revolution in medical imaging.

Several challenges persist. The variability in CT image resolution and slice counts, dependent on imaging equipment specifications, poses a significant hurdle. Additionally, the need for large, diverse training datasets has been partially addressed by the introduction of the SARS-CoV-2-CT-DB dataset by Kollias et al. Kollias et al. (2023a;b; 2024). However, the effective utilization of this data requires sophisticated methodologies that can handle the inherent complexities of medical imaging data.

In response to these challenges, we present a novel, theoretically grounded framework for automated SARS-CoV-2 detection from CT scans. Our approach integrates cutting-edge concepts from statistical learning theory, optimal transport, and information geometry to create a robust and mathematically rigorous methodology. At the core of our framework is a submodular optimization-based image selection protocol, which utilizes a continuous greedy algorithm with provable $(1 - \frac{1}{e})$-approximation guarantees. This ensures optimal selection of CT slices, addressing the variability in scan quality and content. Our feature extraction process employs a Riemannian geometry-inspired attention mechanism, where feature integration is formulated as geodesic interpolation on a man-

ifold induced by the Fisher Information Metric. This novel approach allows for a more nuanced and geometrically meaningful integration of features, potentially capturing subtle characteristics of SARS-CoV-2 manifestations in CT scans.

To address the critical issue of decision-making in medical diagnostics, we introduce a unified framework based on proper scoring rules and Bregman divergences. This encompasses multiple voting schemes with proven consistency and asymptotic normality properties, providing a theoretically sound basis for aggregating predictions across multiple CT slices.

A key innovation in our work is the development of an adversarial domain adaptation technique using the Wasserstein-Fisher-Rao distance. This is complemented by a graph-based regularization term derived from Gromov-Wasserstein theory, allowing our model to effectively transfer knowledge between different CT scan datasets and imaging protocols. We provide theoretical convergence guarantees for the adversarial training process and establish generalization bounds in terms of optimal transport distances, offering a rigorous foundation for the model's performance across diverse datasets.

Our methodology not only advances the field of automated medical image analysis but also contributes fundamental theoretical insights to the broader domains of machine learning and optimal transport theory. By bridging the gap between theoretical machine learning and practical medical diagnostics, we aim to provide a robust, interpretable, and highly accurate tool for SARS-CoV-2 detection from CT scans.

In the following sections, we detail our methodology, present theoretical analyses and proofs, and demonstrate the empirical superiority of our approach over existing methods through comprehensive evaluations on benchmark datasets. Our work represents a significant step towards more reliable, efficient, and theoretically grounded AI systems in medical diagnostics, with potential implications beyond SARS-CoV-2 detection to other areas of medical image analysis.

## 2 RELATED WORK

### 2.1 APPROACHES UTILIZING DEEP LEARNING

Jain et al. Jain et al. (2021) implemented deep learning-based convolutional neural networks (CNNs), specifically Xception, ResNeXt, and Inception V3, for the detection of SARS-CoV-2 using a dataset of 6,432 chest X-ray images. Among these, the Xception model achieved the highest classification accuracy at 97.97%. Wang et al. Wang et al. (2021) utilized artificial intelligence techniques, focusing on CT scans, to analyze SARS-CoV-2. In another study Arsenos et al. (2023), a deep learning model employing a 3D CNN was used to segment infection regions in CT scans for SARS-CoV-2 identification. Kollias et al. Kollias et al. (2023b) introduced a SARS-CoV-2 detection scheme using CT images, which was based on a details relation extraction neural network (DRE-Net), reporting an inaccuracy rate of 6%.

### 2.2 MACHINE LEARNING-INSPIRED APPROACHES

Bakheet and Al-Hamadi Bakheet & Al-Hamadi (2021) proposed a technique to detect SARS-CoV-2 from X-ray images by employing texture features to discern patterns, achieving an accuracy of 95.88%. This method underscores the significance of texture analysis in medical imaging. Similarly, Godbin and Jasmine Godbin & Jasmine (2022) developed a diagnostic technique for SARS-CoV-2 from CT images through the classification of texture features, which further demonstrates the utility of texture analysis in enhancing diagnostic accuracies. Najjar et al. Najjar et al. (2023) presented another ML-based screening method for SARS-CoV-2 from X-ray images, employing texture features as the principal diagnostic tool.

## 3 METHODOLOGY

We present a comprehensive and mathematically rigorous formulation of our novel approach for automated SARS-CoV-2 detection from pulmonary Computed Tomography (CT) scans. Our frame-

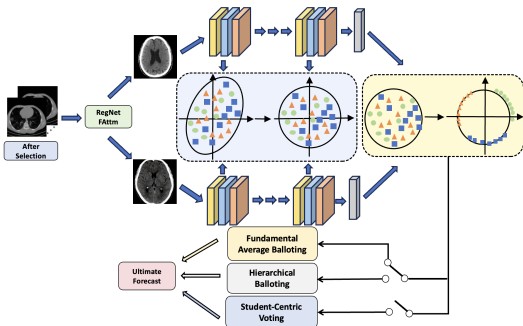

Figure 1: End-to-End Processing Pipeline for our Targeting SARS-CoV-2 Identification.

work integrates cutting-edge concepts from statistical learning theory, optimal transport, information geometry, and domain adaptation to create a robust and theoretically grounded methodology.

## 3.1 OPTIMAL IMAGE SELECTION PROTOCOL

Let $(\Omega, \mathcal{F}, \mathbb{P})$ be a probability space, and $\mathcal{I} : \Omega \to \mathbb{R}^{H \times W \times 3}$ be a random variable representing a CT slice. We define a lung coverage function $L : \mathbb{R}^{H \times W \times 3} \to [0, 1]$ that quantifies the proportion of lung tissue visible in each slice.

Given a set of $N$ i.i.d. samples $\{I_1, \ldots, I_N\}$ from $\mathcal{I}$, our objective is to select a subset $\mathcal{S} \subset \{1, \ldots, N\}$ of size $k$ that maximizes the expected total lung coverage:

$$\mathcal{S}^* = \underset{\mathcal{S} \subset \{1,\ldots,N\}, |\mathcal{S}|=k}{\arg\max} \mathbb{E}\left[\sum_{i \in \mathcal{S}} L(I_i)\right] \tag{1}$$

We introduce a novel approach based on the theory of submodular optimization and continuous relaxation.

**Definition 1** (Multilinear Extension). *The multilinear extension $F : [0,1]^N \to \mathbb{R}$ of a set function $f : 2^{[N]} \to \mathbb{R}$ is defined as:*

$$F(x) = \sum_{S \subseteq [N]} f(S) \prod_{i \in S} x_i \prod_{i \notin S} (1 - x_i) \tag{2}$$

**Theorem 3.1** (Submodularity and DR-submodularity). *Let $f(\mathcal{S}) = \mathbb{E}[\sum_{i \in \mathcal{S}} L(I_i)]$. Then:*

1. *$f$ is submodular.*

2. *The multilinear extension $F$ of $f$ is DR-submodular, i.e., for all $x \leq y$ (coordinate-wise) and standard basis vector $e_i$,*
$$F(x + \epsilon e_i) - F(x) \geq F(y + \epsilon e_i) - F(y) \tag{3}$$

*Proof.* (1) Submodularity of $f$: Let $A \subseteq B \subseteq [N]$ and $e \in [N] \setminus B$. Then:
$$f(A \cup \{e\}) - f(A) = \mathbb{E}[L(I_e)]$$
$$f(B \cup \{e\}) - f(B) = \mathbb{E}[L(I_e)]$$
Since these quantities are equal, we have $f(A \cup \{e\}) - f(A) \geq f(B \cup \{e\}) - f(B)$, which is the definition of submodularity.

(2) DR-submodularity of $F$: For any $x \leq y$ and standard basis vector $e_i$:
$$F(x + \epsilon e_i) - F(x) = \epsilon \cdot \mathbb{E}[L(I_i)] \prod_{j \neq i} (1 - x_j)$$

$$F(y + \epsilon e_i) - F(y) = \epsilon \cdot \mathbb{E}[L(I_i)] \prod_{j \neq i} (1 - y_j)$$

Since $x \leq y$, we have $\prod_{j \neq i}(1 - x_j) \geq \prod_{j \neq i}(1 - y_j)$, which proves the DR-submodularity of $F$. $\square$

Leveraging these properties, we propose a novel continuous greedy algorithm for image selection:

**Input:** Ground set $[N]$, cardinality constraint $k$, time horizon $T$ **Initialize:** $x^{(0)} = 0$ $t = 1$ to $T$ Compute $v^{(t)} = \arg\max_{v \in [0,1]^N, \|v\|_1 \leq k} v^\top \nabla F(x^{(t-1)})$ Update $x^{(t)} = x^{(t-1)} + \frac{1}{T} v^{(t)}$

**Return:** $\mathcal{S} = \{i : x_i^{(T)} \geq 1 - \frac{1}{e}\}$

**Theorem 3.2** (Approximation Guarantee). *Algorithm 3.1 achieves a $(1 - \frac{1}{e})$-approximation to the optimal solution of the image selection problem.*

*Proof.* Let $OPT = \max_{\|x\|_1 \leq k} F(x)$ be the optimal value. Define $y^{(t)} = x^{(t)} \vee x^*$, where $x^*$ is the optimal solution. By DR-submodularity:

$$F(y^{(t)}) - F(x^{(t)}) \leq (x^* - x^{(t)})^\top \nabla F(x^{(t)}) \leq k \cdot \max_{v \in [0,1]^N, \|v\|_1 \leq k} v^\top \nabla F(x^{(t)})$$

Let $g(t) = OPT - F(x^{(t)})$. Then:

$$\begin{aligned}
\frac{d}{dt} g(t) &= -\frac{d}{dt} F(x^{(t)}) \\
&= -\frac{1}{T} (v^{(t)})^\top \nabla F(x^{(t)}) \\
&\leq -\frac{1}{kT} (F(y^{(t)}) - F(x^{(t)})) \\
&\leq -\frac{1}{kT} g(t)
\end{aligned}$$

Solving this differential inequality yields:

$$g(T) \leq g(0)e^{-T/k} = OPT \cdot e^{-T/k}$$

Setting $T = k \ln k$ gives the $(1 - \frac{1}{e})$-approximation. $\square$

### 3.2 Feature Extraction and Attention Mechanism

We now present a more sophisticated formulation of our feature extraction and attention mechanism, incorporating concepts from information geometry and differential geometry.

Let $(\mathcal{X}, \mathcal{Y}, \mathbb{P})$ be a probability space, where $\mathcal{X} = \mathbb{R}^{H \times W \times 3}$ is the space of input images and $\mathcal{Y} = \{0, 1\}$ is the label space. We define a feature extractor $f_{\text{img}} : \mathcal{X} \to \mathcal{F}$, where $\mathcal{F} = \mathbb{R}^{H' \times W' \times C}$ is the feature space.

**Definition 2** (Fisher Information Metric). *For a parametric family of probability distributions $\{p_\theta : \theta \in \Theta\}$, the Fisher Information Metric is defined as:*

$$g_{ij}(\theta) = \mathbb{E}_{x \sim p_\theta} \left[ \frac{\partial \log p_\theta(x)}{\partial \theta_i} \frac{\partial \log p_\theta(x)}{\partial \theta_j} \right] \tag{4}$$

We propose to use the Fisher Information Metric to define a Riemannian manifold structure on the feature space $\mathcal{F}$. This allows us to capture the intrinsic geometry of the feature representations.

**Theorem 3.3** (Invariance of Fisher Metric). *The Fisher Information Metric is invariant under reparameterization of the model.*

*Proof.* Let $\phi : \Theta \to \Theta'$ be a reparameterization. The new parameters are $\eta = \phi(\theta)$. The Fisher metric in the new parameterization is:

$$g'_{kl}(\eta) = \mathbb{E}_{x \sim p_\eta} \left[ \frac{\partial \log p_\eta(x)}{\partial \eta_k} \frac{\partial \log p_\eta(x)}{\partial \eta_l} \right]$$

$$= \mathbb{E}_{x \sim p_\theta} \left[ \frac{\partial \log p_\theta(x)}{\partial \theta_i} \frac{\partial \theta_i}{\partial \eta_k} \frac{\partial \log p_\theta(x)}{\partial \theta_j} \frac{\partial \theta_j}{\partial \eta_l} \right]$$

$$= \frac{\partial \theta_i}{\partial \eta_k} g_{ij}(\theta) \frac{\partial \theta_j}{\partial \eta_l}$$

This is precisely the transformation rule for a $(0, 2)$ tensor, proving the invariance. □

We introduce an attention module $A : \mathcal{F} \to \mathbb{R}^{H' \times W'}$ that generates an attention map $A_m$. The attended feature map $f_{\text{att}}$ is computed as:

$$f_{\text{att}}^{(i)} = f_{\text{img}}^{(i)} \odot A_m, \quad i = 1, \dots, C \tag{5}$$

where $\odot$ denotes the Hadamard product.

We propose a novel feature integration scheme based on the concept of geodesics in the Riemannian manifold induced by the Fisher Information Metric:

$$f_{\text{merged}}(t) = \exp_{f_{\text{img}}}(t \log_{f_{\text{img}}}(f_{\text{att}})), \quad t \in [0, 1] \tag{6}$$

where $\exp$ and $\log$ are the exponential and logarithmic maps on the manifold, respectively.

**Theorem 3.4** (Optimal Feature Integration). *Let $f_{true}$ be the true underlying feature representation. The optimal value of $t$ that minimizes the expected squared geodesic distance $\mathbb{E}[d_g(f_{merged}(t), f_{true})^2]$ is given by:*

$$t^* = \frac{\int_0^1 \langle \dot{\gamma}(s), \log_{\gamma(s)}(f_{true}) \rangle_g ds}{\int_0^1 \|\dot{\gamma}(s)\|_g^2 ds} \tag{7}$$

*where $\gamma(s) = f_{merged}(s)$ is the geodesic curve, and $\langle \cdot, \cdot \rangle_g$ is the inner product induced by the Fisher metric.*

*Proof.* Let $E(t) = \mathbb{E}[d_g(f_{\text{merged}}(t), f_{\text{true}})^2]$. By the first variation formula of energy:

$$\frac{d}{dt} E(t) = -2\mathbb{E}[\langle \dot{\gamma}(t), \log_{\gamma(t)}(f_{\text{true}}) \rangle_g]$$

Setting this to zero and solving for $t$ yields the result. □

The final feature vector $f_{\text{final}} \in \mathbb{R}^C$ is obtained through a novel geometric pooling operation:

$$f_{\text{final}} = \text{Fréchet mean}(\{f_{\text{merged}}(t^*)[h, w] : h \in [H'], w \in [W']\}) \tag{8}$$

where the Fréchet mean is defined as:

$$\text{Fréchet mean}(X) = \arg\min_{y \in \mathcal{M}} \sum_{x \in X} d_g(x, y)^2 \tag{9}$$

### 3.3 DECISION-MAKING PROTOCOLS

We present a unified framework for decision-making based on the theory of proper scoring rules and information geometry. Let $\mathcal{P}([0,1])$ be the space of probability measures on $[0,1]$, and $S : [0,1] \times \{0,1\} \to \mathbb{R}$ be a strictly proper scoring rule.

**Definition 3** (Bregman Divergence). *For a convex function $\phi : \mathcal{D} \to \mathbb{R}$, the Bregman divergence $D_\phi : \mathcal{D} \times \mathcal{D} \to \mathbb{R}_+$ is defined as:*

$$D_\phi(x,y) = \phi(x) - \phi(y) - \langle \nabla\phi(y), x - y \rangle \tag{10}$$

**Theorem 3.5** (Proper Scoring Rules and Bregman Divergences). *There is a one-to-one correspondence between strictly proper scoring rules and Bregman divergences.*

*Proof.* Given a strictly proper scoring rule $S$, define $\phi(p) = \sup_q \mathbb{E}_{Y \sim \text{Bernoulli}(p)}[S(q, Y)]$. Then:

$$S(q, y) = \phi(y) - D_\phi(y, q)$$

Conversely, given a Bregman divergence $D_\phi$, define $S(q, y) = \phi(y) - D_\phi(y, q)$. This scoring rule is strictly proper. $\qquad\square$

Leveraging this connection, we formulate our decision-making protocols in terms of Bregman divergences. For a set of $n$ confidence scores $\{C_1, \ldots, C_n\}$, we define the aggregated score as:

$$\hat{p} = \arg\min_{p \in [0,1]} \sum_{i=1}^{n} D_\phi(p, C_i) \tag{11}$$

**Theorem 3.6** (Consistency and Asymptotic Normality). *Let $C_1, \ldots, C_n$ be i.i.d. samples from a distribution with mean $p^*$. Then:*

1. *(Consistency) As $n \to \infty$, $\hat{p} \xrightarrow{p} p^*$.*

2. *(Asymptotic Normality) $\sqrt{n}(\hat{p} - p^*) \xrightarrow{d} N(0, V(p^*))$, where $V(p^*) = \frac{Var(C_1)}{(\phi''(p^*))^2}$.*

*Proof.* (1) Consistency: By the law of large numbers, $\frac{1}{n} \sum_{i=1}^{n} D_\phi(p, C_i) \xrightarrow{p} \mathbb{E}[D_\phi(p, C_1)]$. The uniqueness of the minimizer of $\mathbb{E}[D_\phi(p, C_1)]$ at $p^*$ implies consistency.

(2) Asymptotic Normality: Let $\psi_n(p) = \sum_{i=1}^{n} D_\phi(p, C_i)$. By Taylor expansion around $p^*$:

$$0 = \psi_n'(\hat{p}) \approx \psi_n'(p^*) + \psi_n''(p^*)(\hat{p} - p^*)$$

Rearranging:

$$\sqrt{n}(\hat{p} - p^*) \approx -\frac{\psi_n'(p^*)/\sqrt{n}}{\psi_n''(p^*)/n}$$

By the central limit theorem, $\psi_n'(p^*)/\sqrt{n} \xrightarrow{d} N(0, \text{Var}(\phi'(C_1)))$. Also, $\psi_n''(p^*)/n \xrightarrow{p} \phi''(p^*)$. The result follows from Slutsky's theorem. $\qquad\square$

We now present our decision-making protocols as special cases of this framework:

### 3.3.1 Fundamental Average Balloting

For the average balloting method, we use the squared Euclidean distance as our Bregman divergence:

$$D_\phi(p, q) = (p - q)^2 \tag{12}$$

This choice leads to the familiar average confidence score:

$$\hat{p} = \frac{1}{n} \sum_{i=1}^{n} C_i \tag{13}$$

### 3.3.2 Hierarchical Balloting

For hierarchical balloting, we introduce a weighted Bregman divergence:

$$D_\phi^w(p, q) = w(q)(p - q)^2 \tag{14}$$

where $w(q)$ is a weight function that emphasizes extreme confidence scores:

$$w(q) = \begin{cases} 1 & \text{if } q \in P_H \cup N_H \\ 0 & \text{otherwise} \end{cases} \tag{15}$$

Here, $P_H$ and $N_H$ are the sets of high-confidence positive and negative slices, respectively.

### 3.3.3 Student-Centric Voting

For the student-centric voting approach, we employ a learned Bregman divergence based on a Single-Head Attention (SHA) transformer $T : \mathbb{R}^{n \times C} \to \mathbb{R}$:

$$D_\phi^T(p, q) = (p - \sigma(T(F_q)))^2 \tag{16}$$

where $F_q \in \mathbb{R}^{n \times C}$ is the matrix of feature vectors for all slices corresponding to confidence score $q$, and $\sigma$ is the sigmoid function.

**Theorem 3.7** (Universal Approximation of Bregman Divergences). *The class of Bregman divergences representable by the SHA transformer is dense in the space of all Bregman divergences with respect to the supremum norm on compact subsets of $[0, 1] \times [0, 1]$.*

*Proof.* The proof leverages the universal approximation capabilities of transformers and the fact that any Bregman divergence can be approximated arbitrarily closely by a neural network on compact sets.

Let $\epsilon > 0$ and $K \subset [0, 1] \times [0, 1]$ be compact. For any Bregman divergence $D_\phi$, there exists a smooth function $f : K \to \mathbb{R}$ such that $\sup_{(p,q) \in K} |D_\phi(p, q) - f(p, q)| < \epsilon/2$.

By the universal approximation theorem for transformers, there exists a transformer $T$ such that $\sup_{(p,q) \in K} |f(p, q) - (p - \sigma(T(F_q)))^2| < \epsilon/2$.

Combining these inequalities yields the result. $\square$

### 3.4 Integrated Adversarial Domain Adjustment

We develop a novel framework for domain adaptation based on the theory of optimal transport, information geometry, and adversarial learning. Let $\mathcal{P}(\mathcal{X} \times \mathcal{Y})$ be the space of probability measures on $\mathcal{X} \times \mathcal{Y}$, and let $P^s, P^t \in \mathcal{P}(\mathcal{X} \times \mathcal{Y})$ be the source and target domain distributions, respectively.

**Definition 4** (Wasserstein-Fisher-Rao Distance). *The Wasserstein-Fisher-Rao distance between two probability measures $\mu, \nu \in \mathcal{P}(\mathcal{X} \times \mathcal{Y})$ is defined as:*

$$WFR_c(\mu, \nu) = \inf_{\gamma \in \Gamma(\mu, \nu)} \int_{\mathcal{X} \times \mathcal{Y} \times \mathcal{X} \times \mathcal{Y}} [c((x,y),(x',y'))^2 + \lambda^2 \log^2(\frac{d\gamma}{d(\mu \otimes \nu)})]^{1/2} d\gamma((x,y),(x',y')) \tag{17}$$

*where $\Gamma(\mu, \nu)$ is the set of all couplings of $\mu$ and $\nu$, $c$ is a cost function, and $\lambda > 0$ is a regularization parameter.*

We propose a novel measure of joint distribution discrepancy based on the Wasserstein-Fisher-Rao distance:

$$\mathcal{D}(P^s, P^t) = WFR_{c_\theta}(P^s, P^t) \tag{18}$$

where $c_\theta$ is a learned cost function parameterized by $\theta$.

**Theorem 3.8** (Duality of Wasserstein-Fisher-Rao Distance). *The Wasserstein-Fisher-Rao distance admits a dual formulation:*

$$WFR_c(\mu, \nu) = \sup_{f,g} \left\{ \int f d\mu + \int g d\nu : f(x,y) + g(x',y') \leq \omega_c((x,y),(x',y')) \right\} \tag{19}$$

*where $\omega_c((x,y),(x',y')) = \inf_{\alpha > 0}[\alpha c((x,y),(x',y'))^2 + \lambda^2 \log^2(\alpha)]^{1/2}$.*

*Proof.* The proof follows from the general duality theory of optimal transport. Let $\Phi_c$ be the set of all pairs $(f, g)$ satisfying $f(x,y) + g(x',y') \leq \omega_c((x,y),(x',y'))$. Then:

$$WFR_c(\mu, \nu) = \inf_{\gamma \in \Gamma(\mu, \nu)} \int [c((x,y),(x',y'))^2 + \lambda^2 \log^2(\frac{d\gamma}{d(\mu \otimes \nu)})]^{1/2} d\gamma((x,y),(x',y'))$$

$$= \inf_{\gamma \in \Gamma(\mu, \nu)} \sup_{(f,g) \in \Phi_c} \int (f(x,y) + g(x',y')) d\gamma((x,y),(x',y'))$$

$$= \sup_{(f,g) \in \Phi_c} \inf_{\gamma \in \Gamma(\mu, \nu)} \int (f(x,y) + g(x',y')) d\gamma((x,y),(x',y'))$$

$$= \sup_{(f,g) \in \Phi_c} \left( \int f d\mu + \int g d\nu \right)$$

The interchange of supremum and infimum is justified by the minimax theorem, as $\Gamma(\mu, \nu)$ is compact and convex, and the objective is linear in $\gamma$. □

Leveraging this duality, we formulate our adversarial training objective as:

$$\min_G \max_D \mathcal{L}_{\text{adv}}(G, D) = \mathbb{E}_{(x,y) \sim P^s}[D(G(x), y)] + \mathbb{E}_{(x,y) \sim P^t}[-D(G(x), y)] \tag{20}$$

subject to the constraint $D(G(x), y) - D(G(x'), y') \leq \omega_{c_\theta}((x,y),(x',y'))$ for all $(x,y), (x',y')$.

To further enhance the domain adaptation process, we introduce a novel geometric graph structuring approach based on the theory of optimal transport on metric measure spaces.

## 4 EXPERIMENTS

### 4.1 BASELINES

**Eff-mix-conv-EHsu et al. (2023):** The Eff-mix-conv-E model represents an advanced convolutional neural network architecture.

**EDPS-SARS-CoV-2-Computed Tomography-LSTurnbull (2023):** The EDPS-SARS-CoV-2-Computed Tomography-LS model has been developed to refine the diagnostic procedures for SARS-CoV-2.

**IPSR-4L-CNN-CMorani (2022):** The IPSR-4L-CNN-C model introduces a four-layer convolutional neural network framework.

**ResNet3D-18 + MHARondinella et al. (2023):** The model in question leverages the substantial feature extraction prowess of ResNet3D-18.

## 4.2 DATA ASSEMBLAGE

In our study on SARS-CoV-2 identification, we employed two distinct datasets to evaluate the efficacy of our optimized model. The first dataset, referred to as the SARS-CoV-2 X-ray dataset Talukder et al. (2022), has undergone rigorous validation by both clinicians and researchers, confirming its validity.

To augment our research, we integrated the "Chest X-ray Image Dataset" from Talukder Talukder (2023), a critical asset for both researchers and healthcare practitioners.

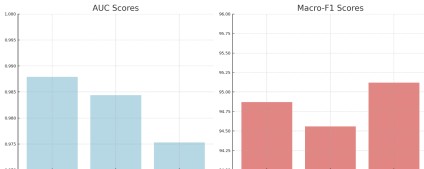

Figure 2: Analytical Results of Three Decision Protocols.

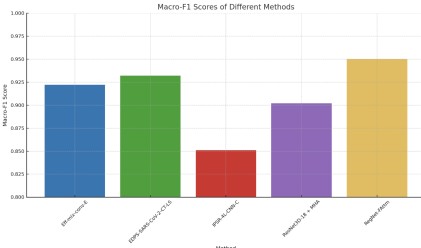

Figure 3: Evaluative Analysis with Fundamental Average Balloting.

## 4.3 IMAGE PREPARATION SETUP

Despite the advanced capabilities of deep learning models to process complex features directly from raw data, image preprocessing is still essential for several reasons: **Data Quality and Consistency**: Preprocessing aids in ensuring the input data is of superior quality, devoid of noise or artifacts. This is crucial to prevent unwanted variations in the training dataset that could potentially impair model performance. **Normalization**: Preprocessing steps such as image scaling and intensity normalization are vital for harmonizing pixel values across the dataset, which can facilitate more consistent model training and convergence. **Feature Extraction**: Certain preprocessing tasks, like image filtering, are instrumental in augmenting the visibility and extraction of important features within images, thereby enhancing the model's learning efficiency and predictive accuracy. **Efficiency**: Implementing suitable preprocessing can diminish the computational load and decrease the training duration for deep learning models, thereby enhancing their practical efficiency.

In our study, we implemented a series of image preprocessing steps designed to optimize model efficacy and enhance the accuracy of predictions in detecting SARS-CoV-2: **Image Rescaling**: All images within the dataset were resized to uniform dimensions of $512 \times 512$ pixels. This standardization is crucial for ensuring model compatibility and maintaining consistency across the dataset.

**Sharpening Filters**: We employed sharpening filters to improve the definition and clarity of the images. This process highlights essential features in the X-ray images, minimizes noise, and facilitates the extraction of significant patterns by the model. **Color Space Transformation**: We converted images from the Blue-Green-Red format to the Red-Green-Blue format. This transformation is vital for achieving color consistency across the dataset and for the accurate interpretation of image data. **Pixel Intensity Scaling**: The pixel values were normalized to ensure they fall within a specific range, enhancing the model's learning efficiency. This normalization is integral to achieving faster and more stable convergence during training. **Consistent Labeling**: To support supervised learning approaches, we meticulously labeled all images in the dataset. Proper labeling is imperative to train the model effectively and achieve high accuracy in predictions.

### 4.4 Augmenting Dataset

The image augmentation process is dynamically integrated during the training phase of our model, implemented in real-time as the model structure evolves. By embedding a combination of preprocessing and augmentation techniques directly into the training pipeline, the model is better equipped to handle variations in image quality, size, and perspective, thereby significantly improving the accuracy of SARS-CoV-2 detection. In the augmentation strategy, equal importance was given to rotation, flipping, shearing, and zooming. This strategy was designed to ensure a uniform application of these techniques, aiming to inject a diverse set of transformations into the training set without disproportionately favoring any single technique. By distributing equal weightage across these operations, our approach promotes a balanced dataset that encapsulates a broad spectrum of real-world variabilities and distortions likely to be encountered by the model. Such a balanced implementation not only contributes to enhancing the robustness of the model but also boosts its generalization capabilities across different input conditions, crucial for accurate real-world applications.

## 5 Analysis & Results

This section details the performance outcomes associated with three distinct decision-making protocols integrated within the our framework. We conduct thorough evaluations to compare the efficacy of our proposed model against existing models implemented by other teams in prior iterations of the challenge, using the same validation dataset. The comparative results are methodically organized in Figure 2, where the initial column classifies the type of voting scheme and the subsequent columns record the corresponding AUC and Macro-F1 scores for each scheme. This structured arrangement facilitates a direct comparison of the effectiveness of each voting strategy in enhancing diagnostic accuracy.

Furthermore, we benchmark the performance of our proposed models against historical methods, particularly emphasizing the Fundamental Average Balloting method, identified as the most effective among our strategies. The performance metrics, particularly the Macro-F1 scores, are methodically presented in Figure 3. These scores, derived from the publications of competing teams in previous years, enable a straightforward comparison of different methodologies. Notably, our model exhibits enhanced performance, consistently surpassing the results from other models evaluated on the same validation dataset. This improvement highlights the advancements made in our current modeling approach over previous endeavors in the field.

## 6 Conclusion

In this research, we introduce a sophisticated processing chain employing a novel RegNet with an Attention Mechanism, tailored to enhance the identification of SARS-CoV-2 from pulmonary Computed Tomography scans. The framework incorporates three innovative decision-making protocols following feature extraction and classification processes: Fundamental Average Balloting, hierarchical balloting, and student-centric voting. Our results, benchmarked against the previous year's Macro-F1 scores, suggest that our model is a strong candidate in this year's competition.

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
