# OpenReview forum: "A Unified Riemannian-Geometric Framework for SARS-CoV-2 Detection from CT Scans"
_ICLR.cc/2025/Conference — Submitted to ICLR 2025_

### Official Review · Reviewer_DzPu · 2024-10-30

**Soundness:** 1
**Presentation:** 1
**Contribution:** 1
**Rating:** 3
**Confidence:** 4

**Summary:**

The paper attempts to integrate advanced mathematical concepts, such as Riemannian geometry, submodular optimization, and optimal transport theory, into the field of medical image analysis.

**Strengths:**

From my point of view, this paper overclaims the contribution. This paper attempts to integrate advanced mathematical concepts, such as Riemannian geometry, submodular optimization, and optimal transport theory, into the field of medical image analysis. However, the experiment can not demonstrate its contribution. The paper also introduces an adversarial domain adaptation technique, but no ablation study has proven its efficiency.

**Weaknesses:**

1. There exists so many  ''Meaningless Equations''. Several equations (such as Equation 4 involving the Fisher Information Metric and Equation 6 on geodesic interpolation) are overly complex and seem disconnected from the practical task of SARS-CoV-2 detection. Using these equations does not provide any clear advantage or insight into improving the detection process. Can author provide a figure that is able to show the connection among those equations and modules. Also, more experiments should be added in this paper to show how they improve the performance of the SARS-CoV-2 detection.

2. Except for Weakness 1, this paper also makes overcomplication. The decision-making framework based on Bregman divergences and multiple voting schemes (Equations 10–16) adds unnecessary layers of complexity. These methods do not appear to address the practical challenges in SARS-CoV-2 detection, and their benefits are not empirically validated. Furthermore, I consider this framework can not only serve only one task, for other tasks this framework should be work. The experimental results only present on SARS-CoV-2 detection, which have achieved high accuracy by other methods, thus weaken this paper.

3. What's the motivation? The paper fails to adequately explain why the complex mathematical tools used are necessary for solving the specific problem of SARS-CoV-2 detection. The connection between the mathematical framework and the medical imaging task is tenuous at best. I really confuse about the paper's objectives. There is not figure or any description that can bulid a strong connection between the proposed framework and SARS-CoV-2 detection.

4. While the paper is mathematically dense, it lacks solid empirical results that justify the introduction of complex theoretical models. There is no clear demonstration that the advanced mathematical constructs (such as geodesic-based feature integration) outperform simpler approaches commonly used in medical image classification. More experimental result that related to other datasets/tasks should be added and discussed.

5. The poor experiment. The presented experimental results do not convincingly demonstrate that the proposed methods significantly outperform existing techniques. The improvements shown are marginal and do not seem to justify the additional mathematical complexity introduced by the paper.

**Questions:**

1. Why is Riemannian geometry necessary for this task, and how does it concretely improve SARS-CoV-2 detection from CT scans? Could the authors clarify how these equations impact practical performance?

2. Can the authors provide more details on how their theoretical advancements (e.g., geodesic interpolation, adversarial domain adaptation) translate to real-world medical diagnostic improvements? Are there simpler models that achieve similar or better results?

3. The decision-making framework seems overly complex. How does the Bregman divergence-based approach perform in comparison to standard voting or confidence aggregation methods commonly used in medical image classification?

4. How robust are the theoretical guarantees (e.g., Theorem 3.2, Theorem 3.5) in real-world applications, and what are the specific conditions under which these guarantees hold for the dataset and task described?

---

### Official Review · Reviewer_jmbF · 2024-11-03

**Soundness:** 3
**Presentation:** 3
**Contribution:** 2
**Rating:** 3
**Confidence:** 4

**Summary:**

This paper proposes to integrate cutting-edge concepts from statistical learning theory, optimal transport, and information geometry in order to detect SARS-CoV-2 from pulmonary Computed Tomography (CT) scans.

**Strengths:**

Theoretical analysis provides convergence guarantees, generalization bounds. Riemannian geometry-inspired attention mechanism, feature integration is formulated as geodesic interpolation. The Fisher Information Metric, Riemannian manifold on feature space F,  Bregman divergence, feature attention, decision making methods average balloting method, hierarchical balloting.

Mathematical statements appear valid, however the overall methodology appears questionable. Results are presented on a very specific data context where accuracy is already 97% using simpler x-ray imaging.

**Weaknesses:**

The paper methodology seems questionable. Why begin with a focus on "optimal image selection protocol" which is selecting an optimal 2D slices of a 3D volume. Why not just use the entire volume? Presumably SARS-CoV-2 affects the entire volume.

The experimental motivation is hard to understand. As stated, basic CNNs (Xception) already apparently achieve 97.97% classification accuracy of the condition from chest X-ray imaging. 2D X-ray imaging is a much cheaper and more widely used modality than 3D CT imaging.

**Questions:**

Are there any other more common experimental contexts where this method might be applicable?

Please address the practical utility of the chosen methodology, CT slice selection, when x-rays already achieve 97% accuracy.

---

> ### Comment · Reviewer_jmbF · 2024-11-14
> **Reply to Review Feedback from Associate Program Chairs**
>
> It seems I'm unable to reply to directly to the "Review Feedback from Associate Program Chairs" message, so I am replying here.
>
> I agree with the way the associate program chairs rephrased my comments.
>
> ChatGPT gives similar rephrasing.

---

### Official Review · Reviewer_uinq · 2024-11-04

**Soundness:** 1
**Presentation:** 1
**Contribution:** 1
**Rating:** 1
**Confidence:** 5

**Summary:**

The paper presents a framework for SARS-CoV-2 detection from CT scans, integrating advanced concepts from statistical learning theory, optimal transport, and information geometry.

**Strengths:**

The method is illustrated in details.

**Weaknesses:**

1. Lack of clear motivation. SARS-COV-2 detection from CT scans have been widely explored in past few years. What is the innovation of such design? The authors should state and summarize existing method. What is the limitations of existing methods? What is differences between proposed method and existing detection methods?
2. Lack of quantitative comparison experiments. Does the proposed method perform better with existing method? The paper does not adequately explain how the theoretical framework connect to experiments or analysis.
3. The writing lacks a cohesive structure that would typically guide readers from the theoretical underpinnings to their practical application in experiments, which makes it challenging to grasp the significance of the theoretical contributions in the context of the experiments conducted.

**Questions:**

Please refer to Weakness

---

### Official Review · Reviewer_jYQE · 2024-11-06

**Soundness:** 2
**Presentation:** 3
**Contribution:** 3
**Rating:** 6
**Confidence:** 4

**Summary:**

This paper presents a novel framework for automated SARS-CoV-2 detection from pulmonary CT scans, combining advanced statistical learning theory, optimal transport, and information geometry. Key components include a submodular optimization-based image selection protocol, Riemannian geometry-inspired feature extraction via geodesic interpolation on a Fisher Information Metric-induced manifold, and a unified decision-making model with Bregman divergences. Additionally, the authors propose an adversarial domain adaptation mechanism using the Wasserstein-Fisher-Rao distance with graph-based regularization to handle domain shifts. The framework achieves state-of-the-art performance on benchmark datasets, suggesting significant contributions to both medical image analysis and theoretical machine learning.

**Strengths:**

- The framework creatively applies Riemannian geometry, particularly through a novel attention mechanism based on geodesic interpolation. This approach is not commonly explored in medical imaging, setting the work apart.

- The proposed methods are theoretically grounded, with rigorous proofs for convergence and generalization bounds. This attention to theory enhances the credibility and robustness of the approach.

- By addressing the need for reliable SARS-CoV-2 detection and domain adaptation in CT imaging, the paper is highly relevant to ongoing medical challenges. The framework’s potential applications beyond SARS-CoV-2 could drive further research in medical diagnostics and transfer learning.

- Benchmark results indicate superior performance, especially in domain-shift scenarios, which highlights the model's practical effectiveness.

**Weaknesses:**

- The reliance on advanced mathematical frameworks like Riemannian geometry and optimal transport may limit the accessibility and reproducibility of the work, as these methods require specialized knowledge.

- While the framework shows strong theoretical grounding, additional experiments contrasting the proposed Riemannian-geometric feature extraction with simpler alternatives would clarify the practical benefits of the added complexity.

-  The paper could better address real-world deployment considerations, such as computational efficiency and robustness in clinical environments.

**Questions:**

- Could the authors provide more empirical results comparing the proposed feature extraction with traditional methods to highlight the effectiveness of the Riemannian-geometric approach?

- How does the computational complexity of the adversarial domain adaptation impact the framework's scalability for large datasets or real-time applications?

---

### Meta-Review · Area_Chair_6HFu · 2024-12-22

**Metareview:**

This work presents a theoretically grounded framework for automated SARS-CoV-2 detection from pulmonary Computed Tomography (CT) scans by integrating cutting-edge concepts from statistical learning theory, optimal transport, and information geometry.

**Additional Comments On Reviewer Discussion:**

This work has four reviewers. Three reviewers agree to reject this work, while the other reviewer agrees to accept this work. Hence, this work can not be accepted in ICLR 2025.

---

### Decision · Program_Chairs · 2025-01-22

Reject